# Stiffness Warming Potential: An Innovative Parameter for Structural and Environmental Assessment of Timber–Concrete Composite Members

Laura Corti * 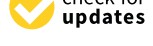 and Giovanni Muciaccia 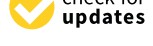

Department of Civil and Environmental Engineering, Politecnico di Milano, 20133 Milan, Italy; giovanni.muciaccia@polimi.it
* Correspondence: laura.corti@polimi.it

**Abstract:** Timber hybridization with concrete is a rising widespread strategy to obtain members with a structural performance comparable to traditional ones—e.g., RC members—but characterized by a greater sustainability potential thanks to the presence of timber-based materials; this solution is of great interest due to its low embodied carbon content, which supports the decarbonization goals set, especially for the building sector. Such systems enhance the concrete and timber favorable properties and ameliorate their detrimental characteristics, both from the structural and environmental perspectives. In general, since these two aspects are generally considered separately, a new parameter is proposed to simultaneously combine a structural performance indicator with a warming potential one. Focusing on composite slabs in bending, the stiffness warming potential ($\lambda$) is introduced, which combines the evaluation of effective bending stiffness (according to Eurocode 5 $\gamma$-method) with the Global Warming Potential—GWP (on the basis of data from Athena Impact Estimator for Building software and data from an Environmental Product Declaration of a timber panel). The method provides a multi-criteria analysis concerning the slab design accounting for vibration, deflection, and acoustic criteria when optimizing the member span. On the other hand, GWP is assessed according to cradle-to-cradle Life Cycle Assessment analysis, where two scenarios with different sustainability levels are encompassed. Results firstly confirm the viability of the novel methodology, with a different outlook on timber–concrete hybrid members, stressing the importance of maintaining thinness of the concrete layer and clearly bringing out the importance of correct re-use and/or a timber recycling management to guarantee effective reductions in terms of $CO_2$ emissions.

**Keywords:** timber–concrete composite (TCC); Life Cycle Assessment (LCA); Global Warming Potential (GWP); effective bending stiffness; sustainability; multi-criteria analysis

## 1. Introduction

The primary concerns of engineers have always been—needless to say—designing structures, managing their construction and maintenance so that minimum safety levels are always guaranteed. Since climate change and related issues became evident topics to be addressed, new challenges arose for engineers: the proper design of a structure cannot exclude actions in favor of sustainability from the very early design phase. When dealing with climate change, the factor more relevant related to the building sector is represented by greenhouse gas (GHG) emissions, contributing to 40% of the total emissions of the sector [1]; other relevant issues are related to land use, the energy efficiency of buildings and the proper management of waste. The urgency to counteract climate change has been intended until recently, as a necessary but rigorous approach, while in recent years the awareness of several opportunities arising from this challenge has created a new, promising research field with practical implications.

The abovementioned step change can be theoretically identified in the development of Sustainable Development Goals by UN in 2015 [2], which have been progressively accepted

and included in everyday life, and concerning specific goals linked with the building sector, the following are of major interest:

- Goal 11—Sustainable Cities and Communities;
- Goal 12—Responsible Consumption and Production;
- Goal 13—Climate Action.

Since these topics may appear abstract, it is fundamental to constantly trace back the discussion to the research and consequently to applications, so the actual challenge is not just to be aware of the existence of such targets, but their practical combination with structural requirements. From this perspective, pursuing innovative solutions is strongly encouraged, from the research on materials to their hybridization and techniques, at different levels of the structure.

A building material that is gaining importance is timber, which is mainly known for its low embodied carbon amount and its renewability, but with relevant structural characteristics: a high strength-to-weight ratio (substantially greater than plain normal concrete and carbon structural steel, the most widespread building materials) [3] and faster manufacturing and construction times [4], with a consequent reduction in heavy vehicle traffic on the building site; moreover, a more readily achievable demolition simplicity leads to better performances in a cradle-to-cradle Life Cycle Assessment (LCA) [5,6], with lower material volumes destined for landfilling. On the other hand, some features of timber exhibit certain disadvantageous characteristics, such as brittleness (in a building, ductility is totally demanded to connections) [7], susceptibility to rot and variability in mechanical properties according to the moisture content [8], with generally higher costs compared to, e.g., traditional RC buildings [9] and excessive deformability [10].

Considering the abovementioned topic of hybridization, it is very common to combine timber with concrete in Timber–Concrete Composites (TCCs), mainly for members such as slabs. Commonly, a concrete topping layer lies over the timber panel, which is generally made of engineered timber, such as Cross-Laminated Timber (CLT), Glued Laminated Timber (Glulam), Laminated Veneer Lumber (LVL), etc.

In a TCC slab, the concrete thickness is designed to resist under compression, while a timber panel resists tensile stresses [4]; in this way, a composite action is achieved, which results in slab optimization, as the cracked concrete portion is noticeably reduced with respect to a reinforced concrete-only case. On the other hand, interface shear forces are transferred by mechanical connectors, which are used to connect the two layers; according to the fastener types, diameters, inclination and timber depth, different resistances and ductility levels are provided [11,12]. A wide range of advantages may be provided by the combined use of timber and concrete:

- The problem of excessive vibration that is generally a great annoyance in mass timber buildings (with only-timber slabs) is resolved by the concrete thickness, which provides additional stiffness. In fact, the low timber bending stiffness (i.e., timber elastic modulus along the main fiber directions is about $^1/_3$ of the concrete elastic modulus) makes it very demanding to fulfill vibration criteria for only-timber long spans, so that in countries where timber is more extensively used as a building material, this issue establishes a prominent research topic in order to first find solutions to avoid such disturbances and then to review the regulations [13];
- Concerning deflection, a very similar issue as the one related to vibration exists, as the addition of concrete topping limits static deflections, ensuring fulfillment of requirements in a serviceability limit state;
- Concerning slab thickness, a thin concrete layer enables longer spans, which can be comparable, e.g., to flat RC slabs or hollow core slabs. Commonly, only-timber slabs cannot reach competitive spans with respect to traditional RC ones due to the fact that generally no more than 7 m spans are viable [14] and due to economic reasons [15], given the considerably higher cost of engineered timber products with respect to concrete;

- Sound insulation is strongly improved with respect to only-timber floors, as due to the concrete density (approximately five times greater than timber one), even a thin concrete layer provides a significant boost of the Sound Transmission Class (STC) [16]; issues in acoustic isolation, especially at low frequencies for lightweight timber floors, are often experienced, even though structural performances are guaranteed [17];
- Slab depth can be reduced up to 50% [18], with significant savings of material (both in the slabs and in the other members due to generally lighter elements), cost-cutting and greater architectural freedom;
- Fire resistance is significantly enhanced, giving better performances with respect to concrete and timber themselves [19], as the low thermal diffusivity of concrete hinders temperature rising in timber, delaying pyrolysis; on the other hand, contrarily to common belief, a timber charred layer ensures a self-protection for the undamaged section, providing an insulating stratum [20];
- Mid- and high-rise timber–concrete hybrid buildings can be designed in regions of medium and high seismicity, while multistorey only-timber buildings hardly result in being competitive with RC and steel equivalent alternatives, as the material volumes required are so large that costs hugely increase [21], besides the negative environmental impact derived from the use of significant quantities of materials. The concrete share provides the necessary lateral stiffness to limit structural and non-structural damages, limiting interstorey drifts caused by lateral loads.
- It is worth underlying that the core of TCC members is the maximum exploitation of the properties of both materials; a clear example lies in timber–concrete hybrid high-rise buildings: timber itself is light but brittle, and so adopted mainly for floors, while reinforced concrete itself exhibits better seismic performance, but it may contribute, in significant terms, to the total building seismic mass. Hence, the use of RC may be limited to the stability system only.
- A separate mention is deserved for the quantification of the environmental impact of TCC members, also thanks to a comparative analysis with equivalent timber-only and concrete-only members. The international standard ISO 14044:2006 [22] for Life Cycle Assessment is adopted as a reference, and the methodology is applied for a comparative analysis of buildings characterized by different materials and structural systems (i.e., timber vs. traditional RC multistorey buildings) [23,24]. According to the aforementioned standard, four different stages should be assessed in an LCA analysis, as detailed in Figure 1, i.e., material production, manufacturing, transport to the building site and construction (stage A), use phase, including both operational impact (energy and water) and interventions to maintain the structure as useable (stage B), building demolition/deconstruction with waste processing (stage C) and re-use, recovery and recycling (stage D). The latter serves to account for benefits and drawbacks out of the traditional building lifecycle.

The actual gap is established by the lack of data on timber–concrete hybrid buildings, where, e.g., slabs beams, and internal partitions are of timber (or the system slabs + beams is a TCC), while columns and shear walls are in reinforced concrete; in this case, the embodied carbon would be significantly lower than in an equivalent RC building, while the weight would be slightly greater than that of an equivalent only-timber solution. Moreover, another experienced shortcoming is the fact that LCA stage D (benefits and loads beyond the system boundary) is often overlooked [25] and—when accounted—a lack of information is detected; when stage D is investigated, it is fundamental to rely on different scenarios [26], mainly for two reasons. A general reason consists in that a unique scenario for an event that is expected to occur after several years contains too many uncertainties; in addition, for timber–hybrid buildings, it should be considered that buildings with engineered timber are new solutions, consider, e.g., that CLT has been on the market from the end of 90 [27], so the oldest buildings are approximately 20–25 years old, and real case studies of End-Of-Life stages are very few.

**Figure 1.** Life-Cycle Assessment stages according to the ISO 14044:2006 standard, reproduced from [22].

Notwithstanding the discussed advantages resulting from hybridization, two aspects require attention. As the first, the fact that timber members, even when satisfying deformability requirements, may easily encounter difficulties in satisfying vibration and acoustic requirements, there is the need of a multi-criteria approach to guarantee people comfort; the same issue is valid for TCC elements, as suggested in [4,18]. It is worth noticing that increasing the concrete thickness—which leads to better acoustic performances—may lead to a reduction in allowable spans [4], since the slab natural frequency is reduced due to a mass increase, and annoying vibrations may be generated. Secondly, a truly synergic combination among the abovementioned multi-criteria structural assessments with environmental requirements from a quantitative perspective is missing; the study presented in this paper suggests a method to concurrently assess these two requirements building a bridge between two fields, which seem so far away in daily design practice. Focusing on the two abovementioned studies—[4,18]—which are characterized by a similar approach to structural performances, the novelty of the proposed research lies in a broader consideration of the environmental perspective, where sustainability requirements are on the same level of importance as structural ones.

The investigation starts by reviewing the influence of connectors on the effective bending stiffness of TCC slabs. The analysis aims to focus on both the effects on structural performances and on the relationship between connector stiffness (and effectiveness) and disassembling simplicity; this last point should be intended as practical and fundamental information for the assessment of the stage D scenario.

Successively, the effective bending stiffness is assessed according to a multicriteria analysis accounting for deformability and vibration requirements to establish the maximum span as a function of the thicknesses of the different layers. Finally, acoustic requirements are checked, increasing the final slab thickness, if required.

Together with multi-criteria analysis, LCA analysis is carried out for each optimized case distinguishing a "sustainable" and a "non-sustainable" End-Of-Life scenario; this is the key step to directing the analysis to a new field of concurrent estimations of effective structural and environmental stiffness, expressed in a unified parameter. In the first case, stage D is not adequately exploited, given that most of the timber portions is delivered to landfilling, while in the second one, an optimized recycling perspective for timber is assessed.

The final step involves a comparative cost analysis, in order to provide an estimation of the differences between a traditional RC flat slab and TCC slab; it should be considered that an equivalent comparison cannot be blindly carried out, as basic differences between two solutions cannot be ignored. The cost analysis is often the main parameter that drives materials and building technique choices, so this study aims to spread the awareness that a new way of thinking—including, as a key parameter, e.g., $CO_2$ emissions—should be pursued.

This paper is organized as follows. The methodology applied to determine the stiffness warming potential parameter and other additional outcomes is explained in Section 2, results and discussion are presented in Section 3 and conclusions with an outlook of future developments is assessed in Section 4.

## 2. Methodology

The effective bending stiffness $(EI)_{eff}$ is the driving parameter to verify the aforementioned criteria and then to link these results with the Global Warming Potential. In order to assess effective bending stiffness, the EN 1995-1-1 Annex B: Mechanically jointed beams [28] procedure is followed, adopting the $\gamma$-method, according to Equation (1).

$$(EI)_{eff} = \sum_{i=1}^{2}\left(E_iI_i + \gamma_iE_iA_ia_i^2\right), \tag{1}$$

where:

- Subscript i specifies the layer, i = 1 = concrete layer and i = 2 = timber layer;
- $E_i$ is the modulus of elasticity;
- $I_i$ is the second moment of inertia, according to Equations (2) and (3). Considering concrete part—Equation (2)—the gross moment of inertia is multiplied by a reduction factor $K_r$ [29] accounting for the cracked section portion, which does not effectively contribute; as for timber, just layers oriented parallel with respect to the span direction are accounted for, so the sum of the considered layers $(t_{2//})$ is used in Equation (3);
- $\gamma_i$ is factor for the efficiency of the mechanical connections, calculated according to Equation (4);
- $A_i$ is the cross-section area;
- $a_i$ is the distance between the centroid of the composite cross-section and the centroids of the i layer, as stated in Equations (5) and (6).

$$I_1 = K_r(= 0.35)\frac{b_1t_1^3}{12}, \tag{2}$$

$$I_2 = \frac{b_2t_{2//}^3}{12}, \tag{3}$$

$$\gamma_i = \begin{cases} 1 & \text{for i} = 2 \\ 1 + \frac{\pi^2E_iA_is_i}{K_iL^2} & \text{for i} = 1 \end{cases}, \tag{4}$$

$$a_1 = \frac{t_1 + t_2}{2} - a_2, \tag{5}$$

$$a_2 = \frac{1}{2\sum_{i=1}^{2}\gamma_iE_iA_i}[\gamma_1E_1A_1(t_1 + t_2)], \tag{6}$$

According to Equations (2)–(6), the following parameters are additionally specified:

- $b_i$ is the cross-section width;
- $t_i$ is the cross-section thickness;
- $s_i$ is the spacing between connectors;
- $K_i$ is the stiffness of the connectors;

- L is the member span.

Values of connector stiffnesses are considered according to references [11,12], which account for wide ranges of solutions. Additional remarks on performance comparisons are provided in Section 3.1.

To assess the effective bending stiffness, it is necessary to set a span length for each slab depth as a first attempt; according to an approach generally adopted in the literature, the following preliminary design criteria is applied, as reported in Equation (7):

$$L = \frac{t_1 + t_2}{0.03},$$ (7)

Considering deflection and vibration requirements, the following criteria are adopted:

- The maximum deflection is determined according to Equation (8), setting it as the maximum allowable deflection $\Delta_{max} = \frac{L}{300}$;
- Vibration requirements are satisfied by complying with a critical length defined in Equation (9), according to [4]. This criterion is chosen for its proven effectiveness, as the formula has been specifically assessed for TCC slabs.

$$\Delta = \frac{5pL^4}{384(EI)_{eff}} \rightarrow L \leq \sqrt[4]{\frac{384(EI)_{eff} \cdot \Delta_{max}}{5p}},$$ (8)

$$L \leq \frac{[(EI)_{eff}]^{0.278}}{4.835 m_L^{0.166}},$$ (9)

$$p = G_1 + G_2 + Q,$$ (10)

According to Equations (8)–(10), the following parameters are additionally specified:

- $\Delta$ is the displacement under the specified load p;
- p is the load acting on the slab, which is a combination of both live loads and the TCC self-weight combination at the Serviceability Limit State (SLS);
- $G_1$ is the structural self-weight (permanent structural load);
- $G_2$ is the permanent non-structural load;
- Q is the live load, assuming residential occupancy;
- $m_L$ is the mass per unit length of a TCC strip of 1 m.

On the other hand, the acoustic requirement is assessed using Equation (11), according to [16], a method that has also been applied to the study presented in [4]. As with the vibration criterion, the selected equation has been exclusively developed for TCC slabs and it is a valid mathematical model when an experimental campaign cannot be carried out or if a preliminary assessment is sought.

$$STC = 20\log_{10}(m_A) + 7dB,$$ (11)

According to Equation (9), the following parameters are additionally specified:

- STC is the Sound Transmission Class;
- $m_A$ is the mass per unit area of the TCC.

STC is especially efficient to estimate noise reduction provided by the structural member when the source is speech sound, and in order to classify and compare efficiencies, a rating is hereby provided, according to the classification suggested in [30] and attached in Table 1.

In case the above discussed requirements are not fulfilled, the procedure is iterated by varying the span automatically in the developed code.

**Table 1.** Sound Transmission Class (STC) rating [30].

| STC Rating | Performance |
| --- | --- |
| STC $\leq$ 30 | Very poor |
| 30 < STC $\leq$ 35 | Poor |
| 35 < STC $\leq$ 40 | Average |
| 40 < STC $\leq$ 45 | Good |
| 45 < STC $\leq$ 50 | Very good |
| STC > 50 | Excellent |

Following this, the evaluation of the environmental burden for each solution is carried out, assessing the Life Cycle Assessment analysis; this step is developed with the support of Athena Impact Estimator for Building (AIE4B) software and an Environmental Product Declaration (EPD) for CLT panels [31]. According to the AIE4B User Manual [32], the End of Life scenario for wood products hardly includes re-use or recycling, which does not represent the most sustainable solution, so an additional scenario is considered according to EPD data; in this second hypothesis, a case study that depicts a real situation as closely as possible is considered, so a 100% recycling scenario is excluded as it is unrealistic to hope that the totality of material from structure dismantling is recycled. Moreover, in the first case, a no-replanting hypothesis for trees is assumed, while in the second one, correct wood management is ensured, and trees are replanted. In both cases, just concrete and CLT materials are considered, and steel for connectors is disregarded; this simplifying hypothesis is adopted due to the fact that the environmental impact of connectors is assumed to be negligible with respect to the connected parts due to their very low volume ratio. Some insights into the role of shear connectors from the mitigation of emissions perspective are however portrayed subsequently. Two hypotheses for CLT panels with their characteristics are presented in Table 2. It is worth specifying that the expected service life of CLT panels is taken as equal to 50 years, according to the reference EPD [31] for the CLT panels.

**Table 2.** End of Life scenarios for CLT panels.

| | Scenario 1—Landfilling | Scenario 2—Recycling |
| --- | --- | --- |
| Software and Tools | AIE4B software | EPD (CLT panel) |
| Wood management | Replantation | No replantation |
| End of Life scenario for timber panels | | |

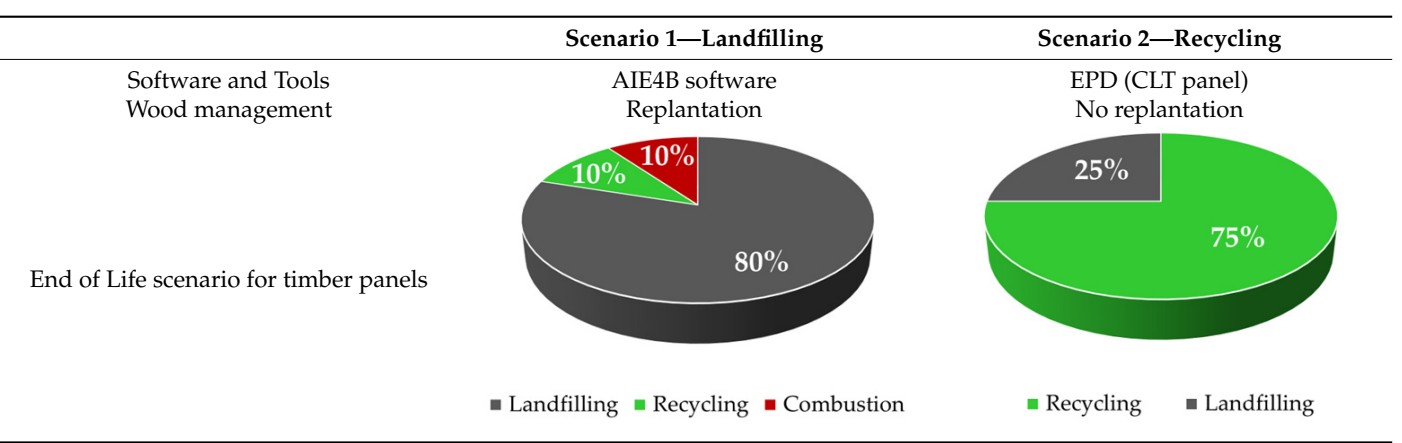

Hence, the Global Warming Potential is estimated summing up the contributions of timber and concrete components, as shown in Equation (12).

$$\text{GWP}_{\text{tot}} = \text{GWP}_{\text{timber}} + \text{GWP}_{\text{concrete}}, \tag{12}$$

Table 3 reports GWP values referring to 1 m$^3$ of concrete C30/37 and CLT CL24, with the differentiation between the two End of Life scenarios for timber.

**Table 3.** Global Warming Potential (GWP) for a reference volume (1 m$^3$) of concrete C30/C37 and CLT CL24.

|  | Concrete C30/37 | CLT CL24 (Landfilling Scenario) | CLT CL24 (Recycling Scenario) |
|---|---|---|---|
| GWP (kg CO$_2$ eq.) | 478.3 | 155.0 | 72.9 |

The key step of the proposed procedure is the introduction of the Stiffness Warming Potential parameter ($\lambda$), as defined in Equation (13).

$$\lambda = \frac{\text{GWP}}{(\text{EI})_{\text{eff}}}, \tag{13}$$

The $\lambda$ parameter suggests an innovative perspective concerning the design of composite elements, where deflection, vibration, acoustic and environmental performances are all simultaneously accounted for; in the case where one parameter does not fulfill a specific requirement, the analysis is repeated until all criteria are satisfied.

The last step of the proposed procedure consists in a comparative cost analysis, in order to provide a general outline of differences according to material thicknesses; in Table 4, costs for 1 m$^3$, respectively, of the C30/37 concrete class and CL24 CLT strength class are outlined. In the listed values, the following costs are enclosed: manufacturing, transport from production to the construction site, workforce and building set up. In the final cost of concrete C30/37, the cost of electro-welded steel mesh with steel grade B450C (mesh 20 × 20 cm, with wire diameter $\Phi$ = 8 mm) is included. When correct wood management that considers replantation, reforestation projects (certified by governments [33]) and the cutting of emissions is ensured [34], cost compensation is applied and a reduction in the CLT panel cost is provided. Costs are estimated on the basis of current Northern Italian market.

**Table 4.** Cost for a reference volume (1 m$^3$) of concrete C30/C37 provided with steel mesh, CLT CL24 and CLT CL24 with the replanting project and discount arising from carbon credits.

|  | Concrete C30/37 | CLT CL24 | CLT CL24 Carbon Compensation |
|---|---|---|---|
| Cost (€/m$^3$) | 180 | 530 | 475 |

In order to check the practicability and the applicability of the proposed parameter, 12 TCC slabs with different thicknesses (and consequently different allowable spans) are analyzed (Table 5), where four only-timber slabs are also considered as benchmarks.

**Table 5.** Studied slabs with their ID and thicknesses of concrete and CLT layers.

| Slab ID | Concrete Thickness—$t_1$ (mm) | CLT Thickness—$t_2$ (mm) |
|---|---|---|
| CLT-1 | 0 | 100 |
| CLT-2 | 0 | 150 |
| CLT-3 | 0 | 200 |
| CLT-4 | 0 | 300 |
| TCC-1.1 | 50 | 100 |
| TCC-1.2 | 100 | 100 |
| TCC-1.3 | 120 | 100 |
| TCC-2.1 | 50 | 150 |
| TCC-2.2 | 100 | 150 |
| TCC-2.3 | 120 | 150 |
| TCC-3.1 | 50 | 200 |
| TCC-3.2 | 100 | 200 |
| TCC-3.3 | 120 | 200 |

**Table 5.** *Cont.*

| Slab ID | Concrete Thickness—$t_1$ (mm) | CLT Thickness—$t_2$ (mm) |
|---|---|---|
| TCC-4.1 | 50 | 300 |
| TCC-4.2 | 100 | 300 |
| TCC-4.3 | 120 | 300 |

Considering loads acting on the slabs the following values are defined:

- $G_1$ is computed according to timber and concrete thicknesses;
- $G_2 = 2.00\frac{kN}{m^2}$
- $Q = 2.00\frac{kN}{m^2}$.

A sketch of the TCC study case is presented in Figure 2a where the arrangement of shear connectors is also outlined: self-tapping screws characterized by a diameter of 12 mm and a total length of 150 mm, with a threaded length of 100 mm, are inclined at $\pm45°$; for every case connector, spacing is constant, with s = 250 mm. According to literature investigations [11,12], the connector stiffness is assumed as $K_S = 60$ kN/mm. In Figure 2, two other examples of connections in the Timber–Concrete composite slabs are also presented.

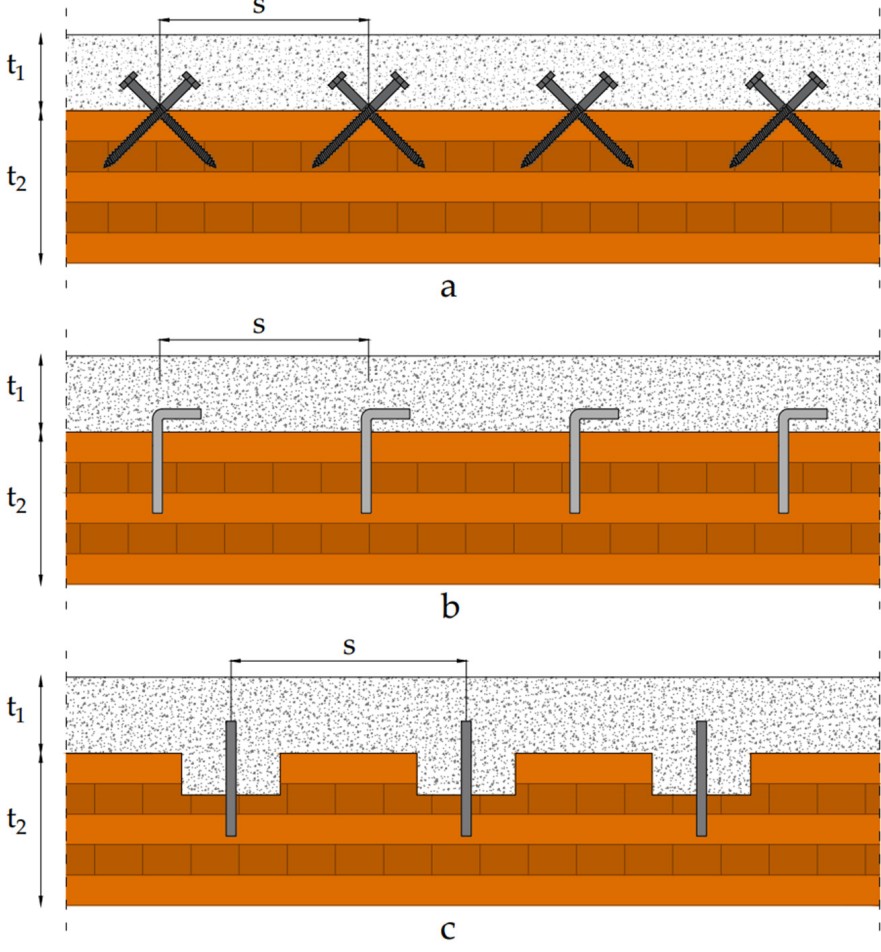

**Figure 2.** (**a**) TCC connection with 45° inclined screws; (**b**) TCC connection with glued steel bars; (**c**) TCC connection with dowels and notches in concrete.

Apart from the abovementioned features, the following parameters concerning geometry and material characteristics are established in Table 6.

**Table 6.** TCC slab characteristics.

|  | **Concrete** | **Timber (CLT)** |
|---|---|---|
| Width | 1000 mm | 1000 mm |
| Thickness | 0, 50, 100, 120 mm | 100, 150, 200, 300 mm |
| Density | 2300 kg/m$^3$ | 490 kg/m$^3$ |
| Young's modulus | 32,800 MPa | 12,000 MPa |

In order to develop the described procedure, a MATLAB script has been implemented, so that iterative steps are facilitated and a wide number of TCC slabs with different geometries can be simultaneously tested. The designed methodology is sketched in Figure 3. According to the procedure, the possibility of studying different scenarios for the same slab is evident; this characteristic should be intended both as a theoretical and practical advantage. From a theoretical perspective, there is the possibility of monitoring the consequences of parameter variation contributions to the creation of a dataset where comparisons are simplified, while from a practical point of view, this tool is useful for design purposes, since different choices are available according to, e.g., a fixed span.

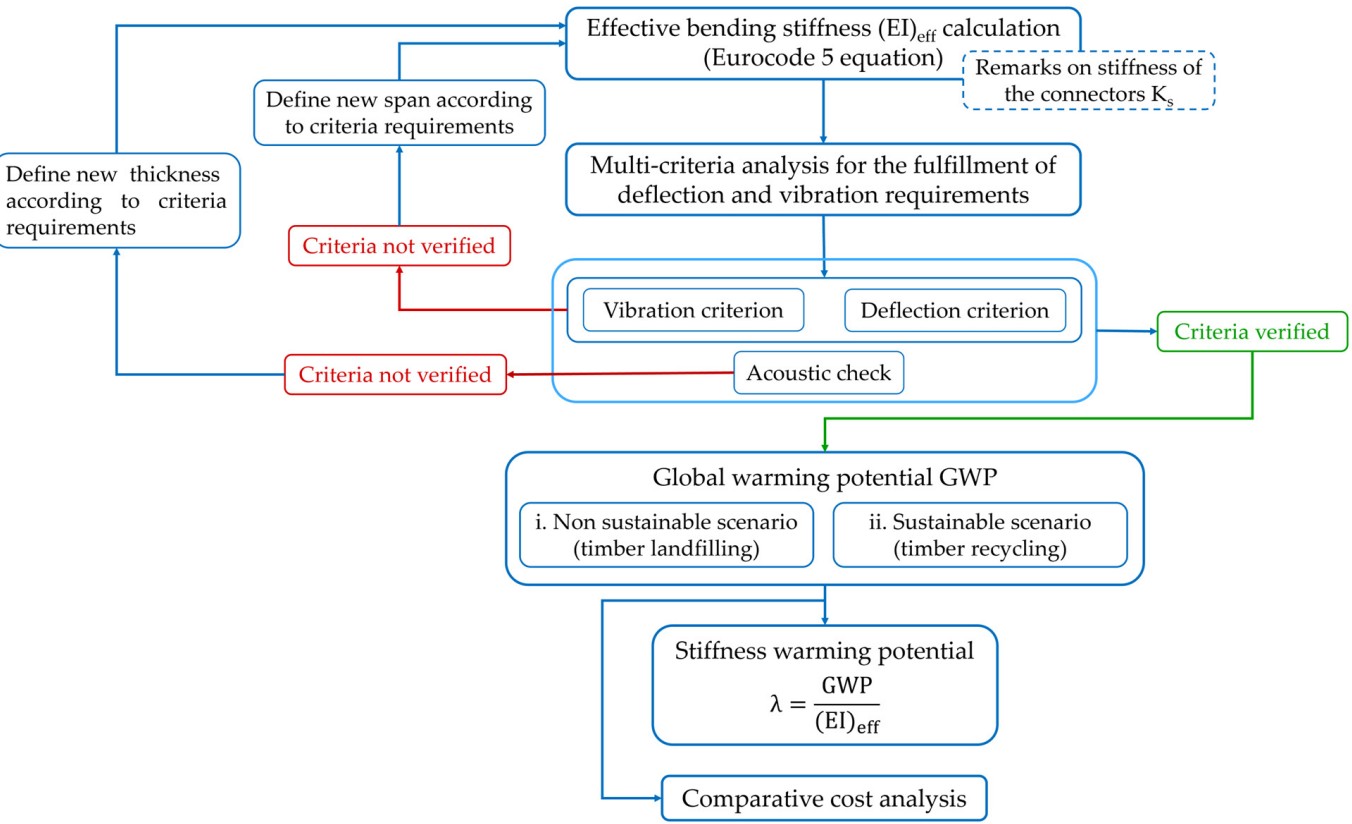

**Figure 3.** Designed methodology to assess Stiffness Warming Potential.

In the next sections, results are analyzed and discussed from the perspective of the optimization of the TCC section.

## 3. Results and Discussion

In this section, results are presented according to the following pattern:

- Section 3.1 presents results and considerations on stiffness of the connectors and their role in the analysis;
- Section 3.2 focuses on the outcomes of the multi-criteria analysis and on difficulties in simultaneously fulfilling the different requirements;

- Section 3.3 presents GWP results, according to the established parameters;
- Section 3.4. presents and discuss results in terms of the Stiffness Warming Potential;
- Section 3.5 shows comparative cost analysis results.

### 3.1. Stiffness of the Connectors and Their Role in the Sustainability Issue

In this investigation, the environmental impact of connectors in terms of GWP has been disregarded, given that their contribution in terms of volumes is considered negligible with a comparison to CLT panels and concrete slabs amounts. However, it is possible to discuss the connectors' role from an environmental perspective since each type of shear connector contributes to the effective bending stiffness of the composite slab with different slip moduli. Figure 4 reports the slip modulus vs. effective bending stiffness of two slabs with a fixed geometry, as specified in Table 7: the only varying parameters are the connector type (and consequently slip moduli—$K_S$—of which values have been adapted from literature [11,12]) and the spacing between connectors, which depends on specific issues linked with the connector. The lowest values correspond to nail or screw connection systems, then intermediate ones correspond to dowel connections—generally vertical with respect to the interface between timber and concrete—and finally dowel connections (commonly inclined of 45°) with notches.

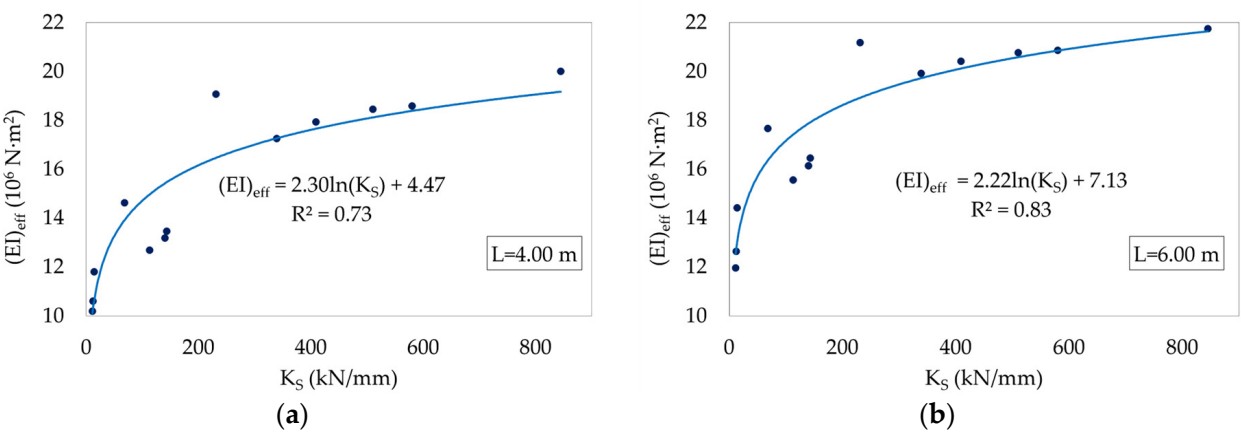

**Figure 4.** Effective bending stiffnesses of two TCC slabs—span length L = 4.00 m (**a**) and L = 6.00 m (**b**)—with different slip moduli according to several alternatives of the shear connection system.

**Table 7.** TCC slab characteristics for an assessment of the relationship between the connector slip modulus and effective bending stiffness.

|  | Concrete | Timber (CLT) |
|---|---|---|
| Slab span | L = 4.00 m and L = 6.00 m | |
| Thickness | 50 mm | 200 mm |
| Width | 1000 mm | 1000 mm |
| Young's modulus | 32,800 MPa | 12,000 MPa |

Even though direct LCA analyses are not carried out, it is possible to focus on how different types of shear connectors may facilitate or hinder structure dismantling; self-tapping screws are simple connection systems that are easily removed and significant portions of the CLT panel are apt to be recycled or even re-used, while on the other hand, notched connections with dowels affect a much greater portion of the slab, so that greater amounts of material may be destined to landfilling or incineration. In this way, the first suggested example fits adequately in a cradle-to-cradle perspective, while the second does not have the simplicity in disassembling that supports a reduction in $CO_2$ emission reductions, within a cradle-to-cradle approach.

### 3.2. Multi-Criteria Analysis

Concerning purely structural requirements, it is interesting to focus on the driving criterion for each slab. As shown in Figure 5, the preliminary design criterion assessed for this study is adequate, as results are generally aligned with vibration and deflection criteria requisites. Considering benchmark pure-timber slabs with shorter spans, they are driven by a deflection criterion, while the longest one (CLT-4) is limited by a vibration criterion. This observation acknowledges the typically noticed vibrational issues of pure-timber long-span slabs. On the other hand, all TCC slabs—except from TCC-1.1 and TCC-1.2 cases—are driven by a vibration criterion, which in case of the longest spans (e.g., TCC-4.1, TCC-4.2 and TCC-4.3), reports a significant reduction with respect to the deflection criterion.

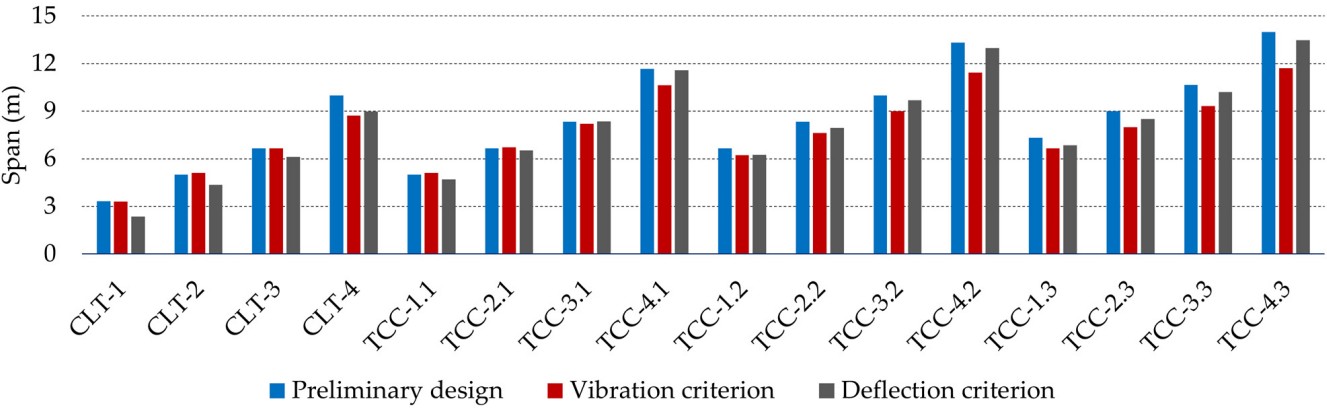

**Figure 5.** Spans according to preliminary design, vibration and deflection criteria.

Considering TCC-1.1 and TCC-2.1, vibration and deflection criteria establish very similar thresholds (respectively 5.11 m and 4.71 m for TCC-1.1, 6.71 m and 6.53 for TCC-2.1), but, ultimately, the deflection requirement is the crucial one; given that the slab is thin with a very limited concrete contribution, the reason for this outcome is attributable to the similarity with CLT-x solutions, where the deflection criterion drives the analysis.

Considering the whole dataset, it is worth focusing on effective bending stiffness values, which, as shown in Figure 6, depend on the total slab thickness, with just slight differences according to the slab layer organization; in the same figure, spans are also represented, with two different clear trends according to the thickness. As for t > 350 mm, an increasing slab depth leads to milder maximum allowable spans. For shorter slabs, the thickness is a more significant parameter, as acknowledged by the slope coefficient of the trendlines.

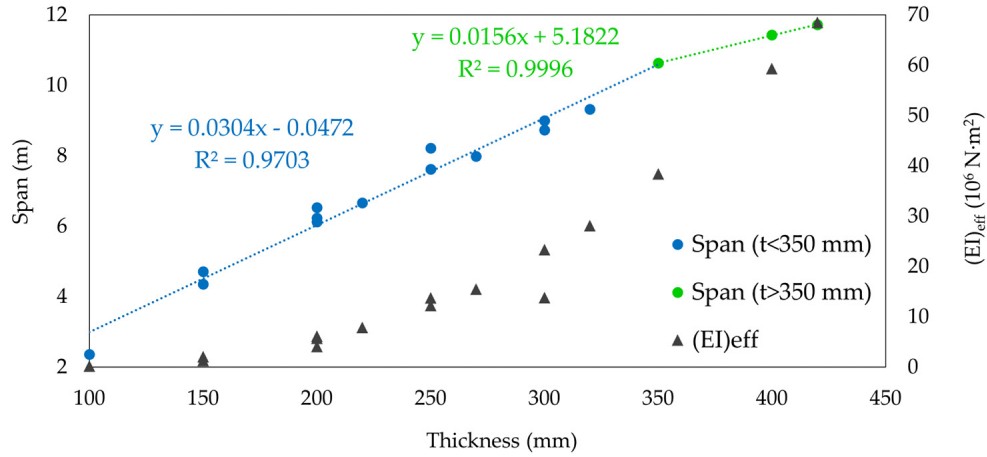

**Figure 6.** Effective bending stiffness and span versus slab thickness.

Considering again the thickness–span relationship, it is necessary to focus on how timber and concrete thicknesses affect slab allowable spans (Figure 7); concrete thickness significantly affects the span for thin slabs, and its influence decreases for thicker depths. Considering, e.g., the curve characterized by $t_t$ = 300 mm in Figure 7b, allowable spans for solutions TTC-4.2 and TCC-4.3 are respectively 11.43 m and 11.72 m, confirming that when switching from a concrete topping of 100 mm to 120 mm, few advantages are recorded. This result has been already obtained [4], but in this case, a more thorough analysis is carried out, given that this is considered a preliminary outcome for the whole study. This result should also be read from the environmental perspective, as a thick concrete layer is not an efficient solution, both from the structural viewpoint (span growth is reduced) and the environmental one (the concrete share of $CO_2$ emissions is substantially greater than timber one). This fact is linked to the vibration criterion fulfilment, as according to Equation (9), the driving parameters are the effective bending stiffness and mass per unit length, so increasing the concrete thickness leads to a moderate $(EI)_{eff}$ increase with a strong growth of $m_L$; on the other hand, increasing the timber thickness brings similar effects on the effective bending stiffness with a more controlled increase in the slab mass.

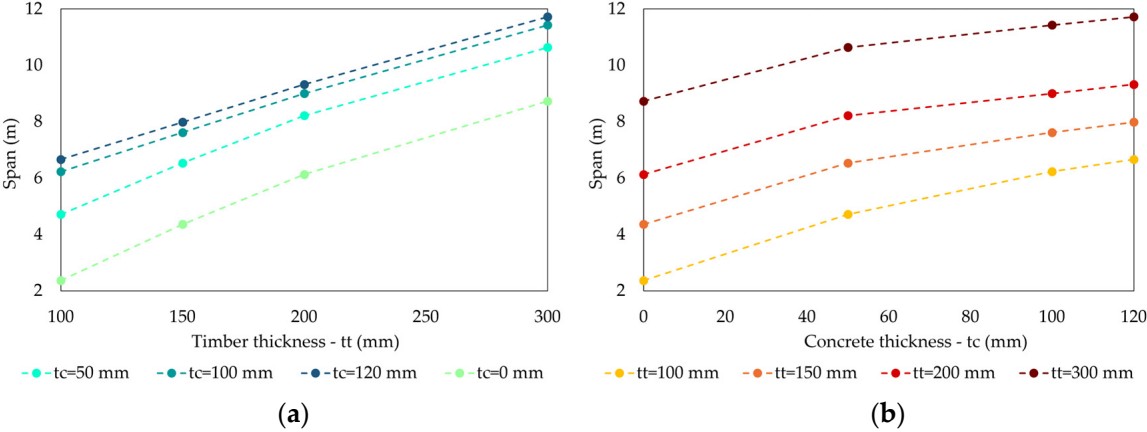

(**a**)  (**b**)

**Figure 7.** (**a**) Timber thickness and span, with data classified by concrete thickness; (**b**) concrete thickness and span, with data classified by timber thickness.

### 3.3. GWP Results

The Global Warming Potential impacts of the analyzed slabs are hereby presented according to the two depicted scenarios. Apart from the quite obvious outcome that increasing the concrete thickness brings an increase in $CO_2$ emissions, it is worth focusing on the comparison of the two scenarios, as the importance of the correct management of materials through the accurate enforcement of recycling methods is highlighted; data on the GWP of concrete and timber are presented in Figure 8.

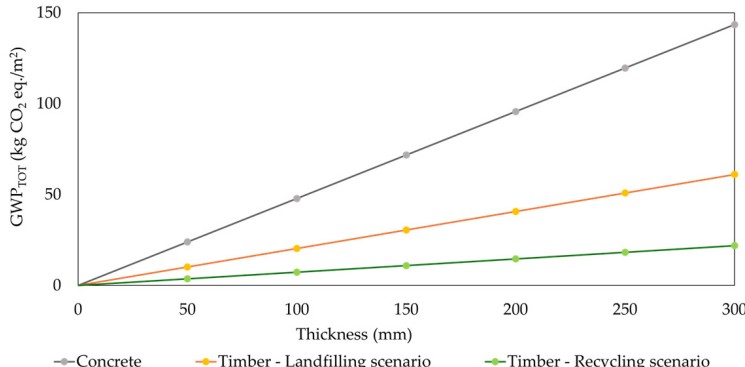

**Figure 8.** GWP impact of concrete and timber. Timber is characterized by two trends, which represent landfilling and recycling scenarios.

It is immediately clear that the concrete thickness strongly affects environmental results, as a mild increase in the concrete topping layer deeply changes the situation, but it is even more important focusing on a comparison of the two scenarios, as considering the recycling hypothesis, an overall reduction in emissions is achieved, as the timber impact in this case is 64% lower than in the "Landfilling scenario". From these early results, the importance of correct timber management is highlighted as the benefits are not confined to the material as such, but they significantly affect the whole TCC member.

In Figure 9, results are presented, and the substantial gap between the two scenarios is immediately evident, so real advantages that come from the recycling hypothesis are confirmed. Therefore, these results should be observed from two perspectives: the first one, by comparing two scenarios and the second one consists of the comparison of the GWP trend according to thicknesses.

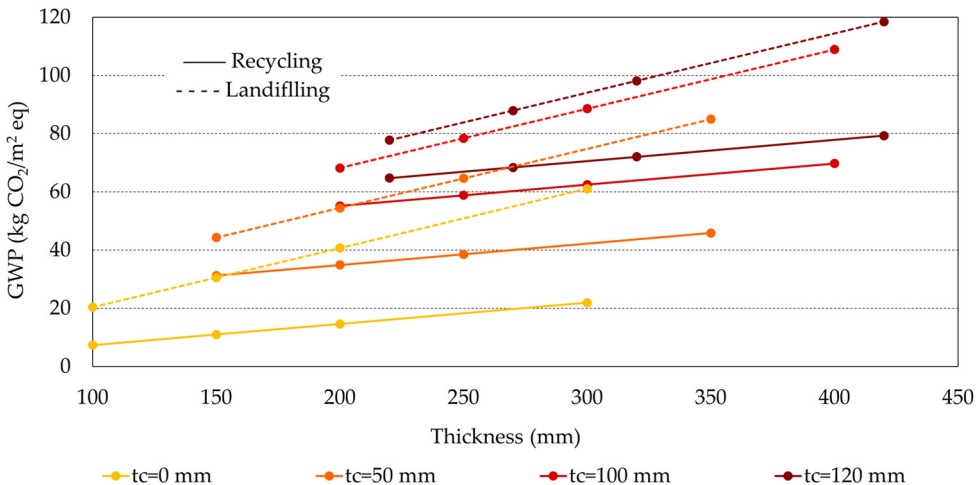

**Figure 9.** Global Warming Potential (GWP) of analyzed slabs according to landfilling and recycling scenarios.

The recycling scenario apports several benefits, as the curve slope is noticeably lower than in the landfilling case; in particular, for longer spans, the landfilling scenario impact is considerably larger, so the correctness of material management becomes even more important. Considering, e.g., the curves characterized by $t_c$ = 50 mm, for the shortest span, the ratio between the recycling and landfilling scenario is 0.70, while for the longest one, it is 0.53; given that concrete is considered to be treated in the same manner in both cases, this difference is all owing to timber divergent management. In case recycling is not considered for timber, the consequences are extremely unfavorable; considering CLT-4 and TCC-4.2, the GWP value of the first one in case of the landfilling scenario is 61.02 kg $CO_2/m^2$ eq., while regarding the second case with the recycling hypothesis, the result is 69.73 kg $CO_2/m^2$ eq.; the contradiction in these results is evident, where the impact of a slab composed of 300 mm of timber and 100 mm of concrete is slightly higher than a 300 mm pure timber solution. This inconsistency is readily solved considering the recycling scenario for CLT-4, as GWP = 21.90 kg $CO_2/m^2$ eq., which is approximately 1/3 of the compared TCC-4.2 slab. Lastly, it is fundamental to recall that the GWP obviously increases when increasing thicknesses and spans—with differences according to the slab composition—but the step forward proposed in this research is to figure out whether it is possible to establish the most efficient structural–environmental combination, according to the λ parameter assessed in Section 3.4.

### 3.4. Stiffness Warming Potential Results

The key outcome of the suggested methodology lies in the derivation of the Stiffness Warming Potential parameter λ, which is the fulfillment of the sought unified consideration of structural and environmental perspectives. Results should be read always bearing in

mind the parameters involved in Equation (13) so optimal values are intended as a balance between low environmental impact and structural performances. Very low values of the λ parameter do not necessarily mean a highly sustainable solution; however, very low values of the GWP brings a reduction in λ, and the same consequence is noticed for slabs with a thick concrete layer where the effective bending stiffness is considerably high. Therefore, it is worth underlying the importance of interpreting results in a critical way with a clear understanding of the whole process.

The proposed approach aims at investigating the most efficient solution, which balances structural and environmental performances; since the needed effective bending stiffness is selected, slab thickness is roughly assessed, with concrete and timber combinations just marginally affecting this results while they deeply impact the GWP. In this context, the notion of the λ parameter develops, and the first focus is specifically on a comparison of the two assessed scenarios—landfilling and recycling—presented in Figure 10.

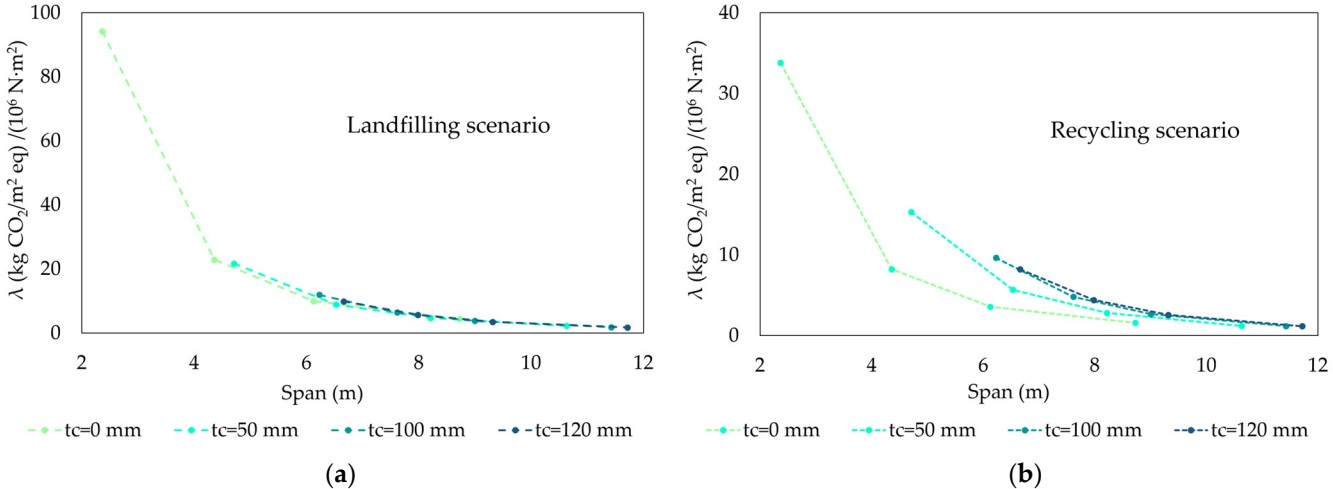

**Figure 10.** (**a**) Slab span and Stiffness Warming Potential—landfilling scenario; (**b**) slab span and Stiffness Warming Potential—recycling scenario.

Focusing on Figure 10a especially for medium and long spans, there are no tangible differences according to the slab composition: in this case, the λ parameter exclusively depends on the span. These results should be read bearing in mind that the landfilling scenario is assessed, so just 10% of timber is destined to be recycled, while the greatest share (80%) is delivered to landfilling; in this situation, the presence of timber is practically useless from the λ parameter perspective, as whichever combination is chosen, the same Stiffness Warming Potential is achieved. Thereby, data represented in Figure 10a can also potentially be represented with a unique fitting curve, which again underlines the ineffectiveness of the timber non-recycling hypothesis: very different combinations of concrete and timber in the slab are characterized by the same structural–environmental performance, which could lead to the decision to thicken concrete portions in order to control costs.

On the other hand, considering Figure 10b, the results are differentiated, and the timber recycling protocol is acknowledged; since the span is chosen, different λ parameters can be obtained according to the assessed combination. This result strongly affects the design, as given that effective bending stiffness is guaranteed roughly by the total thickness, the decision can be taken according to environmental considerations, as in this case, substantial differences are recognized. Nevertheless, it is worth underlying that this gap becomes lesser for the longest spans, in the case of beams where the major effective bending stiffness is needed. Above all, given that recycling scenario data are not adequately fitted by a single curve, there is an additional remark on the importance of material combinations.

The remarks emerging from Figure 10 may also be accounted for from the perspective of slab thickness. Figure 11 reports the same results but clustered by concrete thicknesses.

The curves characterized by $t_c$ = 100 mm and $t_c$ = 120 mm do not experience significant differences according to timber thickness increases, with just mild discrepancies in the λ parameter; this observation becomes even more evident for thicker spans where for all considered $t_c$ values, the Stiffness Warming Potential ranges between 1.16 and 1.58 kg $CO_2$ eq./($10^6$ N·m$^2$).

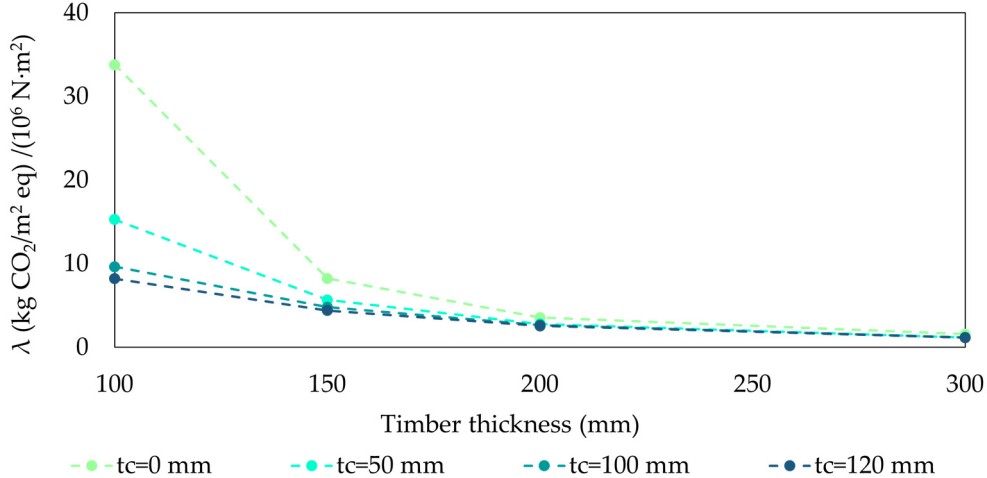

**Figure 11.** Timber thickness and Stiffness Warming Potential results, with data classified by concrete thickness. The figure only considers a recycling scenario.

Results should be read by also considering the necessary thickness for slab spans; since the span has been established, a preliminary thickness value is fixed (or the opposite, the procedure is the same), and consequently, the effective bending stiffness is assessed, and layers are precisely defined. In this way, the slab GWP is calculated, and finally, the λ parameter is established, and comparisons with other solutions are carried out. Layer arrangement is undoubtedly an important characteristic, but precise thicknesses can be adjusted according to the desired performance; as a confirmation of this, (EI)$_{eff}$ values corresponding to the same slab depth with different layer thicknesses reported in Figure 6 are very close one to another. Therefore, the importance of the λ parameter is thereby highlighted as playing a more prominent role: when two slab results are similar based on comparing effective bending stiffness outcomes, the choice should be influenced by environmental considerations, so this new parameter acts as a decision-making support tool.

It can be concluded that the discussion regarding the potential use of the λ parameter has proven the effectiveness of the proposed approach; starting from the question of whether it was possible and meaningful to define a new parameter to simultaneously account for structural criteria and the GWP impact, the obtained results confirm that this procedure is not only feasible but seems to be significantly valuable in supporting a sustainable building design.

### 3.5. Cost Analysis

The end results obtained within the designed methodology concern an analysis of slab costs in a comparative framework, given that apart from the standard solution, an additional one that includes cost reduction thanks to carbon credits is considered. TCC slab costs are outlined in Figure 12, where a slight cost reduction is observed for the second solution.

It is worth underlying the differences between the two situations are not massive given that firstly manufactured timber is characterized by some fixed costs, which at present cannot be avoided and secondly due to the fact that the carbon credit market is a very recent business, which is recently developing at an outstanding rate, but it still needs to be comprehensively considered. In this framework these results should be considered

as a promising potential future research field where cost differences between CLT and concrete—which now are remarkable—will be considerably reduced.

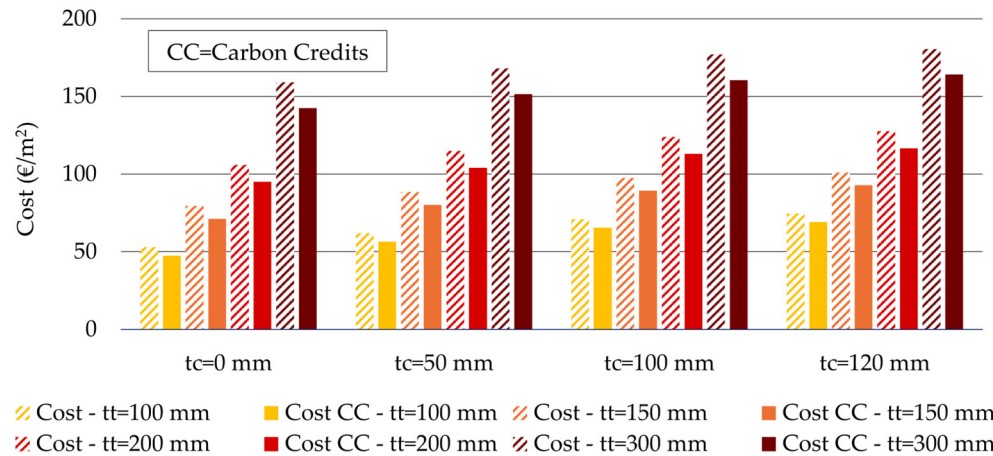

**Figure 12.** Cost analysis. Standard scenario with full costs and a discounted scenario arising from carbon credits are presented.

## 4. Conclusions

One of the major acknowledged issues in modern building design is the absence of polluting emission quantification, coupled with the fact that, in case, e.g., the GWP impact is evaluated, this step is provided at the end of the design phase. Consequently, a fundamental key item is missed: environmental considerations can no longer be considered as an additional, or even worse as an optional, step after structural design. Differently, the need to provide indicators or parameters for use during the structural design phase is urgent. In this context, the idea of introducing a Stiffness Warming Potential for members in bending, has grown so as to provide a coupled assessment of structural and environmental performances; this strategy aims at concurrently developing calculations to the effect that the GWP impact is a key driving parameter and not a marginal value barely or "afterwards" considered.

The Stiffness Warming Potential, or $\lambda$ parameter, is introduced and applied in the case of TCC slabs; it addresses the environmental impact per the effective stiffness unit, and it can be adopted as a design parameter to immediately quantify and compare the GWP impact of different solutions. It emerges that a thin concrete layer leads to a more efficient TCC slab with respect to an equivalent one with a thicker concrete thickness only as long as correct timber management, in terms of both of recycling and replating, is guaranteed. Differently, the same combination of the slab depth and span can be achieved basically, independently of the combination of concrete and timber thicknesses.

The observed benefits of adopting a composite slab that is "properly managed" range from environmental advantages to the fact that lighter slabs are more favorable from a seismic design viewpoint as they contribute to a lower seismic mass; moreover, vibrational requirements are more easily fulfilled with a lighter slab, with respect to similar effective bending stiffness. A thin concrete layer is efficacious by meeting satisfying acoustic requirements, and where this criterion is already fulfilled, provides an appreciable enhancement of acoustic isolation.

Some future developments concerning the developed methodology are foreseen: from the perspective of the mitigation of emissions, research on a "Recycling scenario" would lead to even more efficient recycling percentages; this development is directly linked to the cost analysis, as the use of recycled timber is planned, and further cost reduction is expected. In this context, the cost analysis, which in this early stage is assessed as an additional step, will become another key driving parameter to support the decision-making process favoring TCC slabs instead of traditional solutions. One of the most restrictive driving

parameters nowadays is the substantially higher cost of engineered wood in comparison to, e.g., concrete. However, an increasing awareness of the significant advantages provided by such solutions is expected to increase the general interest toward them and to potentially improve the policies related to the carbon credits market.

Finally, another future development, which can be expected, is accounting for concrete recycling; today concrete recycling is already a reality, which is applied in a widespread way, but quantification of its impact in the context discussed in this paper is still missing. This last future step should be taken to be part of a mechanism, which is continuously moving forward, and novelties are always considered as challenges whence innovative solutions act as a bridge between the population of our day and sustainable development. The proposed tool was adopted preserving its original flexibility, which lies in the facility to adopt additional criteria or weights different to the ones already introduced, while maintaining the proposed methodological approach.

It can be concluded that the key feature of this approach is a close connection with the real world, where the sustainability of all aspects of human life is now beginning to be perceived as a crucial point; in this context, it is fundamental that research on civil engineering supports this trend, given that the building sector holds a primary role in the emission of pollutants.

**Author Contributions:** Conceptualization, G.M.; Methodology, L.C.; Investigation, L.C.; Writing—original draft, L.C.; Writing—review & editing, G.M.; Supervision, G.M. All authors have read and agreed to the published version of the manuscript.

**Funding:** This research received no external funding.

**Institutional Review Board Statement:** Not applicable.

**Informed Consent Statement:** Not applicable.

**Data Availability Statement:** Not applicable.

**Conflicts of Interest:** The authors declare no conflict of interest.

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
