# Peer review of "Stiffness Warming Potential: An Innovative Parameter for Structural and Environmental Assessment of Timber–Concrete Composite Members"

_sustainability, doi:10.3390/su152014857_

Round 1

Reviewer 1 Report

The manuscript deals with the stiffness warming potential for the structure and environmental assessment of the timber-concrete composite member. Overall, the research work is effective to assess the material based on the structure design and global warming potential. However, there are various writing concerns. I have the following major concerns.

1.       Abstract structure should be improved: The first statement of the abstract must be improved. I recommend adding the background of the problem in a more formal way.

2.       Grammatical Mistakes in Abstract e.g., comma is required “deflection, and acoustic criteria.”, “GWP”, CLT, and EPD abbreviations not explained before, no need to write most of the important words with capital letters e.g., “Stiffness Warming Potential”, etc.

3.       The results are not well represented at the end of the Abstract.

4.       Keywords are the important points of the research work: there are 7 keywords: it must be 5-6. Nothing is given in the abstract about the “Sustainability” and “life cycle assessment”.

5.       The first character of the bullet is always capital.

6.       Few paragraphs are very big, and a few are small. Especially at the end of the Introduction section and in the conclusion, there are many single-statement paragraphs that disturb the concentration of the reader.

7.       You discussed stage D, what are stages A, B, and C. Better, if graphically illustrated?

8.       There are too many grammatical mistakes in the whole manuscript. I recommend correcting all.

9.       You should also add the contributions of your research work as compared to [4] and [18], as you mentioned they have also used the same factors for the reduction of allowable spans.

10.   The methodology and results are well represented using graphical content. 

Too many mistakes. I recommend an extensive check. 

Reviewer 2 Report

Work on following commnets:

1) re-write abstarct to includes novelty part of the work, present few crucial results too

2) How typical parameter, say for effective bending stiffness are considered? add justification in methodology section

3) is it possible to provide some cross validation study, recommended to add

4) conclusion can be added with future scope

Round 2

Reviewer 1 Report

Dear Authors,
The article is based on evaluating structural and environmental assessment of timber concrete composite members as compared to the concrete in the structure of the buildings. It is an interesting topic and has economic and environmental benefits in the construction industry. I have a couple of serious concerns. Please consider them.

1.       I will just start with your Abstract. Your argument in the first line of the abstract is “Timber hybridization with concrete is a rising strategy nowadays and consists of good structural performance and greater sustainability as compared to traditional ones”, but my view contradicts this claim. First, it is also required to discuss the type of structures you considered in your research work e.g., big buildings, roads, bridges, homes, etc., where the application of the CLT is better than the concrete C30 because each type requires highly different strength and performance. Secondly, sustainability is not only related to carbon emissions, recycling, landfill saturation, or life cycle assessment goals, but it must also require the sustainability of the material you are recommending. I see there is no analysis or simulation given in your work to analyze the expected life of the CLT, as we know that from our experience the concrete life is more than 25-30 years for home structure. The building material is experiencing a wide range of internal and external stress due to the environment, and as a result, the material life is affected. Maybe CLT is better in structure performance for a few of the indicators or maybe for small homes only, but overall, it is not possible to claim without experiments or maybe some simulations are required to analyze that.

2.       You analyzed the design of the building for acoustic, deflection, vibration, and environmental performance, but missing the most significant parameters i.e., the expected life of the structure. In my view, claiming to use the CLT instead of concrete C30 may provide poor circumstances for the firms without experimenting. Yes, the research work has had various important analyses related to global warming potential, thickness, material properties, stiffness, cost, etc. This can be a conceptual work to have future studies and concise results related to the given analysis. 

A minor English check is required.

Round 3

Reviewer 1 Report

Dear Author,

It is interesting to see you cited previous research work about the performance (especially life expectancy) of products having timber hybridization with concrete further comments. No further comments. 

I think the English language quality is good.